# The Mycovirome in a Worldwide Collection of the Brown Rot Fungus *Monilinia fructicola*

**DOI:** 10.3390/jof8050481

**Published:** 2022-05-06

**Authors:** Rita Milvia De Miccolis Angelini, Celeste Raguseo, Caterina Rotolo, Donato Gerin, Francesco Faretra, Stefania Pollastro

**Affiliations:** Department of Soil, Plant and Food Sciences, University of Bari Aldo Moro, 70126 Bari, Italy; ritamilvia.demiccolisangelini@uniba.it (R.M.D.M.A.); celeste.raguseo1@uniba.it (C.R.); r.caty@tiscali.it (C.R.); donato.gerin@uniba.it (D.G.); stefania.pollastro@uniba.it (S.P.)

**Keywords:** mycovirus, stone fruit, +ssRNA virus, mitovirus, splipalmivirus, botourmiavirus, benyvirus, hypovirus, fusarivirus, barnavirus, parvovirus

## Abstract

The fungus *Monilinia fructicola* is responsible for brown rot on stone and pome fruit and causes heavy yield losses both pre- and post-harvest. Several mycoviruses are known to infect fungal plant pathogens. In this study, a metagenomic approach was applied to obtain a comprehensive characterization of the mycovirome in a worldwide collection of 58 *M. fructicola* strains. Deep sequencing of double-stranded (ds)RNA extracts revealed a great abundance and variety of mycoviruses. A total of 32 phylogenetically distinct positive-sense (+) single-stranded (ss)RNA viruses were identified. They included twelve mitoviruses, one in the proposed family *Splipalmiviridae*, and twelve botourmiaviruses (phylum *Lenarviricota*), eleven of which were novel viral species; two hypoviruses, three in the proposed family *Fusariviridae*, and one barnavirus (phylum *Pisuviricota*); as well as one novel beny-like virus (phylum *Kitrinoviricota*), the first one identified in Ascomycetes. A partial sequence of a new putative ssDNA mycovirus related to viruses within the *Parvoviridae* family was detected in a *M. fructicola* isolate from Serbia. The availability of genomic sequences of mycoviruses will serve as a solid basis for further research aimed at deepening the knowledge on virus–host and virus–virus interactions and to explore their potential as biocontrol agents against brown rot disease.

## 1. Introduction

Mycoviruses are viruses infecting a broad range of hosts in the major taxa of fungi and oomycetes, including many important pathogens of plants, insects, and humans [1,2,3,4], mycorrhizal fungi [5], yeasts [6], and mushrooms [7,8].

Recent studies have greatly enhanced our knowledge of the diversity and evolution of fungal viruses [9]. Mycoviruses are characterized by either single- or multi-segmented genomes that can be naked or differently packaged in virus-like particles (VLPs) with an isometric, spherical, bacilliform, or filamentous shape [10]. The majority of known mycoviruses possess linear double-stranded RNA (dsRNA) or positive-sense (+) single-stranded RNA (ssRNA) genomes [2,11], although mycoviruses with linear negative-sense (-) ssRNA [12] as well as single-stranded DNA (ssDNA) genomes [13] have been reported. Thanks to the advent of high-throughput next-generation sequencing (NGS), the number of mycoviral sequences available in the GenBank database has drastically increased to >770. According to the International Committee on Taxonomy of Viruses (ICTV; https://talk.ictvonline.org/taxonomy, accessed on 20 January 2022), mycoviruses are currently classified into 23 viral families and one unclassified viral genus [6]. However, taxonomic changes and the establishment of new families are very frequent due to the fast discovery of novel species that remain unclassified or are classified into taxa not yet formally approved [14,15].

Most mycoviruses lack an extracellular phase in their replication cycle and can be transmitted, with various efficiencies, horizontally to other fungal individuals via hyphal anastomosis, and vertically to asexual or sexual spores [1,16]. It is known that horizontal transmission may be hindered by vegetative incompatibility, a type of non-self-recognition that often induces programmed cell death and restricts the spread of molecular parasites [17]. This can pose limitations to the use of mycoviruses as biocontrol agents against fungal diseases. However, the strength of the barrier to mycovirus spreading due to vegetative incompatibility can vary considerably [18]. For instance, free virions of ssDNA viruses within the family *Genomoviridae* can infect fungal hosts on artificial media or in plant tissues [19,20], as well as mycophagous insects [21]. Moreover, some viruses, such as Sclerotinia sclerotiorum mycoreovirus 4 (SsMYRV4), can suppress vegetative incompatibility in a fungal host [22].

Recent studies on mycoviruses revealed fundamental aspects of the complex virus–host interactions that can be influenced by environmental conditions, host strain, temperature, and possible co-infecting viruses, resulting in various effects on their hosts [23]. Multiple viral infections are common in fungi and different viruses may co-exist and replicate in a single host [7,24,25]. Complex virus–virus interplays in co-infected hosts can be synergistic, antagonistic, or neutral interactions and have practical outcomes for the potential use of mycoviruses as biocontrol agents (e.g., [26]).

Mycovirus infections are mostly cryptic or responsible for mild symptoms with no apparent effects on fungal biology [27]. Nevertheless, some mycoviruses are associated with a range of altered phenotypic traits, beneficial or detrimental to their hosts [1,2]. Killer phenotypes in yeasts represent a classical example of a beneficial viral–host interaction [28]. Mycoviruses can increase the virulence (hypervirulence) of plant- and human-pathogenic fungi [29,30]. On the opposite, mycoviruses associated to host debilitation may cause reduced virulence (hypovirulence), impaired growth and sporulation, abnormal morphology and pigmentation, as well as altered metabolic regulation [1,2,31,32,33]. In mushroom cultivations, viruses are associated with economically important diseases causing heavy yield losses in several fungal species, such as *Agaricus bisporus*, *Lentinula edodes*, and *Pleurotus* spp. [7,34,35]. In pathogenic fungi, mycoviruses can cause hypovirulence and might be exploited as potential biocontrol agents against fungal diseases [18]. Moreover, mycoviruses might play a role in the adaptation of fungi to certain ecological niches. For instance, Pestalotiopsis theae chrysovirus-1 (PtCV1) modulates endophytic and phytopathogenic traits of *Pestalotiopsis theae* [36]; Curvularia thermal tolerance virus (CThTV), in a three-way symbiosis, confers heat tolerance to both the endophyte *Curvularia protuberata* and its host plant, a tropical panic grass, and allows both organisms to grow at high soil temperatures [37].

The most well-known example of successful biocontrol mediated by a mycovirus is the application of Cryphonectria hypovirus 1 (CHV-1) against the chestnut blight fungus *Cryphonectria parasitica* [38]. The widespread success of CHV-1 led to extensive studies of mycovirus-mediated hypovirulence, resulting in the discovery of several mycoviruses with biocontrol potentials in important phytopathogenic fungi, such as the white mould fungus *Sclerotinia sclerotiorum* [13], *Ophiostoma novo-ulmi* [39] causing the Dutch elm disease, the grey mould fungus *Botrytis cinerea*, the rice blast fungus *Pyricularia oryzae* [40], *Plasmopara halstedii* causing downy mildew on sunflower [41], and soil-borne pathogens, such as *Rhizoctonia solani* [42] and *Rosellinia necatrix* [43].

NGS technologies have changed the approach of investigations on fungal-associated viruses, allowing the detection of single or very few viral sequences in libraries from total RNA [44,45] or small RNA (sRNA) [46,47], although the first approach usually provides a more complete characterization of the fungal virome [48].

*Monilinia fructicola* (G. Winter) Honey is among the most destructive fungal pathogens of rosaceous fruit crops, mainly stone fruits. It induces American brown rot, causing blossom and twig blight and fruit decay, both in the field and postharvest, resulting in heavy yield losses. Following its introduction into Europe [49], the pathogen spread widely and rapidly, and for its greater fitness often prevails over the endogenous species *Monilinia laxa* and *Monilinia fructigena* [50,51].

There is only limited information available on mycoviruses infecting species of the genus *Monilinia*. Earlier studies have revealed the presence of viral dsRNAs and VLPs in *M. fructicola* isolates from peaches and nectarines in New Zealand [52], with no apparent effects on virulence, grow rate, and sporulation. Tran et al. [53] identified a hypovirus (Sclerotinia sclerotiorum hypovirus 2), a betapartitivirus (Fusarium poae virus 1), and a mycoflexivirus (Botrytis virus F) in *M. fructicola* and *M. laxa* isolates from Western Australia. Coinfection of the three mycoviruses in *M. fructicola* increases its grow rate in vitro by 10% but does not affect its virulence on fruits. The complete genome of a novel virus, Monilinia umbra-like virus 1 (MULV1), infecting *M. fructigena* was recently obtained by total RNA sequencing of mummified peach fruits [54]. Recently, Jo et al. [55] reported 20 partial and uncharacterized genome sequences of mycoviruses obtained by in silico shotgun assembly of Illumina RNA-Seq reads from 18 libraries of *M. fructicola*, *M. fructigena*, and *M. laxa* [56]. Most of them were from two Italian *M. fructigena* isolates.

The aim of the present work was to investigate the mycovirome of *M. fructicola* in a worldwide collection of isolates of the fungus from different hosts using an NGS-based metagenomic approach.

## 2. Materials and Methods

### 2.1. Fungal Isolates and Growth Conditions

A collection of fifty-eight isolates of *M. fructicola* from various host plants and locations worldwide was used (Table 1). All isolates, stored at −80 °C in glycerol (10% *v*/*v*) until use, were revitalised and grown on potato dextrose agar (PDA; infusion from 200 g peeled and sliced potatoes kept at 60 °C for 1 h, 20 g dextrose, adjusted at pH 6.5, 20 g agar Oxoid No. 3, per litre of distilled water), at 21 ± 1 °C in the dark for 7 days to obtain conidia. Aliquots (10 mL) of conidial suspension (0.7–1.3 × 10^6^ conidia mL^−1^), obtained by scraping the plate surface with water with a few drops of Tween 20 added, were added to 190 mL of complete medium (CMV; 10 mL solution A (10 g KH_2_PO_4_, 100 mL^−1^ distilled water), 10 mL solution B (20 g NaNO_3_, 5 g MgSO_4_·7H_2_O, 0.1 g FeSO_4_, 100 mL^−1^ distilled water), 1 mL micro-nutritive solution [57], 10 g yeast extract Difco, 10 g casein acid-hydrolyzed Difco, 10 g bacto peptone Difco, 20 g glucose, and 1 mL vitamins stock solution [58] filtered and added to media after sterilization at 121 °C for 20 min) and maintained 60 h in an orbital shaker (150 rpm) at 25 ± 1 °C in the dark. Mycelium was collected on Miracloth (Calbiochem, San Diego, CA, USA) under a slight vacuum and washed twice with sterile distilled water.

### 2.2. dsRNA Extraction and Purification

*M. fructicola* isolates were grouped in twelve pools, each of 4–5 isolates, basically according to their origin (Table 1). dsRNAs were isolated from each pool according to [59], with some modifications. In detail, a mix of an equal amount (2.5 g) of mycelium from each isolate was freeze dried and ground to a fine powder with a mortar and pestle in liquid nitrogen. To each pooled sample, 20 mL of GPS buffer (0.2 M glycine, 0.1 M Na_2_HPO_4_, 0.6 M NaCl, pH 9.5) were added and the mixture was stirred on ice for 30 min. Then, 2 mL of 10% SDS (sodium dodecyl-sulfate), 0.2 mL of β-mercaptoethanol, 20 mL of phenol pH 8.0, and 20 mL of chloroform-pentanol (25:1) were added and the mixture gently shaken for 30 min and then centrifuged at 9000 rpm for 20 min at 4 °C. The aqueous phase was recovered and extracted again with 20 mL of phenol and 20 mL of chloroform-pentanol (25:1). Afterwards, it was added with ethanol to a final concentration of 15% *v*:*v* and 1 g of CF11-cellulose (Whatman, Kent, UK), and stirred for 15 min. Nucleic acids adsorbed on cellulose were precipitated by centrifugation at 4500 rpm for 15 min at 4 °C. The pellet was resuspended with 10 mL of 15% ethanol STE (0.1 M NaCl, 0.05 M tris-HCl, 0.001 M Na_2_EDTA, pH 8.0) and used for the chromatographic column preparation by using a vacuum manifold (Supelco Visiprep^TM^, Bellefonte, PA, USA). After washing with 100 mL of STE added with 15% ethanol, nucleic acids were finally eluted with 10 mL STE and overnight precipitated with ethanol (1:2, *v*:*v*) at −20 °C. The pellet obtained after centrifugation at 14,000 rpm for 30 min at 4 °C was finally suspended in 200 μL of TE (0.01 M Tris-HCl, 0.001 M Na_2_EDTA, pH 8.0).

Nucleic acids were digested with 0.02 U mL^−1^ DNAase I (Life Technologies Corporation, Carlsbad, CA, USA) and then with 500 ng mL^−1^ ribonuclease A from bovine pancreas (Sigma-Aldrich, St. Louis, MO, USA), following the manufacturer’s instruction. Proteins were removed sequentially with equal volumes of phenol:chloroform-isoamyl alcohol (1:1, 24-1) and chloroform, and then dsRNA was pelleted at 14,000 rpm for 20 min after addition of 0.1 vol. 3 M sodium acetate (pH 5.5) and 2.5 vol. ethanol at −80 °C for 30 min. The pellet, washed with cold 70% ethanol, was vacuum-dried and re-dissolved in 15 μL of nuclease-free water.

The presence and quality of dsRNA were checked by electrophoresis on 1% (*w*/*v*) agarose (Certified^TM^ Molecular Biology Agarose, Bio-Rad Laboratories, Hercules, CA, USA) gel run in TAE buffer (0.04 M tris-base, 0.02 M CH_3_COOH, 0.001 M Na_2_EDTA, pH 8.0) at 90 V for 110 min. The quality and quantity of dsRNA were assessed by absorption spectra using a NanoDrop 2000 Spectrophotometer (Thermo Scientific Inc., Wilmington, DE, USA). Prior to usage, dsRNA samples were also quantified using the Qubit RNA BR Assay (Invitrogen, Paisley, UK).

### 2.3. dsRNA-Seq Library Preparation and High-Throughput Sequencing

cDNA libraries were prepared from 0.8 µg dsRNA extracts from each pool of *M. fructicola* isolates using the TruSeq^TM^ RNA Sample Preparation Kit v2 (Illumina, San Diego, CA, USA) in accordance with the manufacturer’s instructions, with some modifications. After validation by spectrophotometric (NanoDrop 2000) and fluorometric (Qubit^®^ 2.0) assays and agarose gel electrophoresis, normalized cDNA libraries were multiplexed (12 libraries per lane) and custom-sequenced at the Genewiz Inc. NGS Service (South Plainfield, NJ, USA), using an Illumina HiSeq2500 platform with a 2 × 150 bp configuration and an average insert size of ~200 bp.

### 2.4. Bioinformatics Analysis

After a quality check using FastX-tools (http://hannonlab.cshl.edu/fastx_toolkit; accessed on 27 September 2020), the reads were trimmed and assembled de novo using CLC Genomics Workbench 7.0.3 (CLC Bio, Qiagen, Hilden, Germany) with the default settings. SeqMan Pro (Lasergene v.15.0.1; DNASTAR, Inc., Madison, WI, USA) was used for merging the smaller contigs into larger scaffolds. The consensus sequences were then subjected to BLASTN and BLASTX analysis against viral nucleotide and protein sequences in the NCBI database (www.ncbi.nlm.nih.gov; accessed on: 17 November 2020), with the default parameters and the E-value cut-off of 10^−5^. These sequences were filtered for redundancy at a 90% nucleotide identity over 90% of the length, and representative mycoviral sequences were selected from each pool. Selected sequences were classified according to the hit with the highest bit score and used in the subsequent analyses. To determine and correct assembly errors, row reads were mapped back to the de-novo assembled viral contigs, using SeqMan NGen and reads alignments visualized in SeqMan Pro. Frequencies of detection for each reconstructed mycovirus in each pool of isolates, expressed in RPKM (Reads Per Kilobase per Million mapped reads), were quantified using CLC Genomics Workbench. 

Prediction of the open reading frames (ORFs) into the assembled viral genomes was performed using SeqBuilder (Lasergene v.15.0.1; DNASTAR). Protein molecular weight was calculated using the tool available at https://www.bioinformatics.org/sms/prot_mw.html (accessed on: 11 December 2021). Conserved protein domains were identified using MOTIF search (GenomeNet, Japan; https://www.genome.jp/tools/motif) in PFAM, NCBI-CDD, PROSITE pattern and profile databases (accessed on: 14 December 2021).

### 2.5. Phylogenetic Analysis

The amino acid sequences of the replication-associated proteins in *M. fructicola* viruses, the three closest homologues identified by BLASTP analysis, as well as the reference viruses for the family were used for multiple alignments with MUSCLE implemented in MEGA-X [60]. Phylogenetic trees were computed by MEGA-X using the maximum likelihood (ML) method set to the default parameters. Statistical support for the branches was evaluated using a bootstrap analysis [61] with 1000 replicates. iTOL v6 [62] was used to display, annotate, and manage the phylogenetic trees.

### 2.6. PCR Validation of Assembled ssDNA Viral Contigs

To confirm the presence of a putative ssDNA virus detected from pool No. 4, total DNA was extracted from each isolate in the pool using Gentra Puregene tissue kits (Qiagen, Milan, Italy), according to the manufacturer’s instructions. A PCR primer pair (p111Fw 5′-AGTGCCCAAATAGACGTCTCT-3′/ p2573Rv 5′-GCTGCATCGTGTTCTTTTGC-3′) was designed to generate a 1.67 kb product covering the genomic region included between the two available contigs. PCR products were assessed using gel electrophoresis and Sanger sequencing by external service (Genewiz, Takeley, UK).

## 3. Results

### 3.1. Extraction of dsRNA and Analysis of Electrophoretic Profiles

The total amount of dsRNA ranged from 2.2 μg (pool 8) to 20 μg (pool 11). The dsRNAs yielded on agarose gel 1–4 bands having sizes ranging from 1.6 to 3.0 kb for most dsRNA samples, except for the pool No. 5 that showed an additional band of about 11 kb (Appendix A).

### 3.2. High-Throughput Sequencing and Reconstruction of Viral Genomes

A total of 340,001,023 paired-end (PE) reads were generated by Illumina NGS resulting in ~98 Gb of raw sequencing data with a mean quality score (QS) of 36.7. The number of non-redundant reads in the dsRNA samples ranged from 12,863,610 (pool No. 3) to 26,697,938 (pool No. 4) (Table 2).

The assembly of the high-quality reads (QS ≥ 30) yielded an average of about 43,000 contigs per pool, corresponding to 34.1 Mb and 81.9–98.6% of the mapping reads (Table 3). The assembled contigs included 61–91% of small sequences with length < 1 kb (data not shown). An average number of 1339 contigs per pool showed similarity to viral proteins in the BLASTX search and were selected as putative mycoviral sequences. After filtering for redundancy, the number of identified sequences was reduced to an average of 193 per pool, ranging from 67 (pool No. 9) to 588 (pool No. 6).

Viruses from different pools showing identity at the amino acid level >95% were considered variants of the same virus; after manual inspection of alignment, the sequence of each virus with the highest read coverage and sequence length was selected as the representative one.

A total number of 33 viral genomes were identified from the assembled sequence reads (Table 4). Based on sequence analysis, mycoviruses were classified in different taxonomical groups, including 32 viruses that were predicted to represent +ssRNA viruses belonging to seven different viral families within the realm *Riboviria*, phyla *Lenarviricota* (*Mitoviridae*, the proposed family *Splipalmiviridae*, *Botourmiaviridae*), *Pisuviricota* (*Hypoviridae*, the proposed family *Fusariviridae*, *Barnaviridae*), *Kitrinoviricota* (*Benyviridae*), and only one ssDNA virus with similarity to viruses of the *Parvoviridae* family (phylum *Cossaviricota*, realm *Monodnaviria*).

#### 3.2.1. *Mitoviridae* and *Splipalmiviridae*

Twelve sequences showed similarity to viruses in the family *Mitoviridae* (Table 4). The size of the reconstructed sequences ranged from 2298 to 2816 nt, likely representing near-complete genomes, and contained a single ORF encoding an RdRp protein with a conserved domain (PF05919) and length ranging between 683 and 794 aa (77.47–80.84 kDa), except Monilinia fructicola mitovirus 11 (MfrcMV11) which showed a larger protein of 866 aa (99.99 kDa). The highest sequence identity of the RdRp sequence (89.7%) was found between MfrcMV7 and MfrcMV8, while the remaining viruses showed an identity at the amino acid level of ≤47%. Then, they included 11 distinct species, according to the demarcation criteria established by ICTV for mitoviruses (https://talk.ictvonline.org; accessed on: 10 January 2022). Most shared amino acid sequence identities ranging between 63.3% and 99.1%, with mitoviruses of *B. cinerea* (MfrcMV1, MfrcMV9 and MfrcMV12) or *S. sclerotiorum* (MfrcMV2 to 4, MfrcMV6 to 8, and MfrcMV10 and 11), while Monilinia fructicola mitovirus 5 (MfrcMV5) showed low identity (42.3%) with Ophiostoma mitovirus 6. 

A putative new narna-like virus with two genomic segments (RNA1 and RNA2) was found and named Monilinia fructicola splipalmivirus 1 (MfrcSPV1). RNA1 was 2466 nt in length and coded for a single protein 785 aa in size and 89.2 kDa, while the partial sequence of RNA2 was 2123 nt in length and included an incomplete ORF coding for a protein 698 aa in size. RNA1 showed 77.2% aa sequence identity to Erisiphe necator associated narnavirus 13 and about 50% to Plasmopara viticola lesion associated narnavirus 4 and Alternaria tenuissima narnavirus 1; both segments showed aa sequence identity of 53.7% (RNA1) and 47.9% (RNA2) with Botrytis cinerea binarnavirus 2 and other ‘binarnaviruses’ found in *B. cinerea* [14], now included in a group of viruses for which alternative names of polynarnaviruses and splipalmiviruses have been proposed [5,63]. Like other viruses in this group, e.g., [63], MfrcSPV1 showed the RdRp palm subdomains split into the RNA1 (motifs F, A, and B) and RNA2 (motifs C and D) segments (data not shown).

The phylogenetic relationships of the *M. fructicola* mycoviruses and their relatives in the *Mitoviridae* and the proposed family *Splipalmiviridae*, within the phylum *Lenarviricota*, are shown in Figure 1. The phylogenetic analysis grouped mitoviruses into two major clades and 5 subclades and the *M. fructicola* viruses were distributed among them. MfrcSPV1 was grouped with Erysiphe necator associated narnavirus 13, Botrytis cinerea binarnavirus 2, and Plasmopara viticola lesion associated narnavirus 4 in a strongly supported clade much closer to *Mitoviridae* than to *Narnaviridae*.

#### 3.2.2. *Botourmiaviridae*

Twelve viral genomes showed similarity to members of the family *Botourmiaviridae*. The genus *Ourmiavirus* includes plant viruses with a tripartite genome, while the other genera include viruses infecting fungi (*Botoulivirus*, *Magoulivirus*, *Penoulivirus* and *Rhizoulivirus*) or both plants and fungi (*Scleroulivirus*), with a monopartite +ssRNA genome and a single ORF encoding an RNA-directed RNA polymerase (RdRp). 

The sequences from *M. fructicola* ranged from 2382 to 3004 nt in length and contained complete ORFs (645–780 aa; 70.88–89.14 kDa) with an RdRp conserved domain (PF05919). Three RdRp sequences showed high similarity with botourmiaviruses already known in *Monilinia* species. In details, Monilinia fructicola botourmiavirus 5 (MfrcBOV5) showed 99.7% aa sequence identity with Monilinia ourmiavirus B from *M. fructicola*; Monilinia fructicola botourmiavirus 9 and 10 (MfrcBOV9 and 10) showed 86.6% and 85.9% aa sequence identity, respectively, with two viruses identified in *M. fructigena*, Monilinia ourmiavirus I, and Monilinia ourmiavirus A/Mfrg. Four other sequences of Monilinia fructicola botourmiaviruses showed significant levels of similarity (70.3%-86.6% identity) with the RdRp proteins to botourmiaviruses identified in *S. sclerotiorum* (MfrcBOV1 and MfrcBOV3) and *B. cinerea* (MfrcBOV2 and MfrcBOV4). 

According to the ICTV species demarcation criterium (RdRp identity ≤ 90%) within the family *Botourmiaviridae*, MfrcBOV1 to 4, 9, and 10 should represent new species of botourmiaviruses. The remaining viral genomes, Monilinia fructicola botourmiavirus 6, 7, 8, 11, and 12 (MfrcBOV6, 7, 8, 11 and 12), shared sequence identities lower than 55% with botourmiaviruses identified in other fungi and oomycetes. According to the genera demarcation criteria (RdRp identity ≤70%) these should therefore represent members of five new genera in the family *Botourmiaviridae* [64].

The phylogenetic analysis showed that the family *Botourmiaviridae* was subdivided into two major clades and six subclades, each representing a genus. Five of the identified Monilinia fructicola botourmiaviruses were clustered in the genus *Scleroulivirus* (MfrcBOV1, 6, 7, 8, and 12), four in the genus *Penoulivirus* (MfrcBOV4, 9,10, and 11), two in the genus *Botoulivirus* (MfrcBOV2 and 3), and one in the genus *Magoulivirus* (MfrcBOV5) (Figure 2).

#### 3.2.3. *Hypoviridae*

Two large genomic sequences showing high homology with viruses of the family *Hypoviridae* were identified and tentatively named Monilinia fructicola hypovirus 1 and 2 (MfrcHV1 and 2) (Table 4). The family includes just one officially recognized genus, *Hypovirus*, comprising non-encapsidated viruses with +ssRNA genomes of 9.1–12.7 kb that possess one or two ORFs [65]. 

MfrcHV1, with a genome length of 9338 nt and a single ORF of 8754 nt coding for a polyprotein of 2918 aa and 331.49 kDa, showed high aa sequence identity with Sclerotinia sclerotiorum hypovirus 7 (97.7%) and Botrytis cinerea hypovirus 1 (96.9%). The polyprotein showed three conserved domains, an RdRp (PF00680), a PPPDE putative peptidase (PF05903), and a C-terminal helicase (PF00271). 

MfrcHV2, with a genome length of 15,037 nt and a single ORF of 14,355 nt coding for a polyprotein of 4784 aa and 539.32 kDa, showed high aa sequence similarity with Monilinia hypovirus D (93.2%) and Sclerotinia sclerotiorum hypovirus 2 (85.3%). Two conserved domains were detected in the polyprotein, superfamilies 1 and 2 ATP-binding type-1 (PS51192), and C-terminal (PS51194) helicases.

Recently, it has been proposed to replace the genus *Hypovirus* in the family *Hypoviridae* with the two genera *Alphahypovirus* and *Betahypovirus* (e.g., [66]). Hu et al. [67] proposed a third genus, *Gammahypovirus*. Phylogenetic analysis based on the full-length aa sequence of the polyproteins (Figure 3), clustered MfrcHV1 with the type species Cryphonectria hypovirus 3 and Cryphonectria hypovirus 4, Sclerotinia sclerotiorum hypovirus 7, and Botrytis cinerea hypovirus 1 in the proposed genus *Betahypovirus*; MfrcHV2 was clustered with Sclerotinia sclerotiorum hypovirus 2 in the proposed genus *Gammahypovirus* including also Botrytis cinerea hypovirus 5, Sclerotium rolfsii hypovirus 1, and Monilinia hypovirus D.

#### 3.2.4. *Fusariviridae*

Two near-complete viral genomes, provisionally named Monilinia fructicola fusarivirus 1 and 2 (MfrcFV1 and 2), and a partial genome of Monilinia fructicola fusarivirus 3 (MfrcFV3), all related to members of the proposed family *Fusariviridae*, were reconstructed. These consisted of +ssRNA genomes of 7425 nt (MfrcFV1), 7330 nt (MfrcFV2), and 5550 nt (MfrcFV3), each containing two ORFs, the largest ORF1 encoding a polyprotein containing an RdRp conserved domain (PF00680) and superfamilies 1 and 2 ATP-binding type-1 (PS51192) and C-terminal (PS51194) helicases, and the ORF2 encoding a hypothetical protein.

The MfrcFV1 polyprotein (1661 aa, 191.55 kDa) revealed an aa sequence identity of 81.2% to Monilinia fusarivirus G from *M. fructigena* and ~40% with other fusariviruses. The ORF2-encoded protein (504 aa, 57.74 kDa) revealed an aa identity of 71.5% to Monilinia fusarivirus G. It contained a conserved DUF1322 domain (PF07032). 

The MfrcFV2 polyprotein (1667 aa, 192.23 kDa) revealed aa sequence identity (97.4%) to Monilinia fusarivirus B from *M. fructigena* and ≤ 52.9% to other fusariviruses. The ORF2-encoded protein (509 aa, 56.73 kDa) revealed aa identity of 95.5% identity to Monilinia fusarivirus B, with no conserved domains. 

The partial MfcFV3 polyprotein (1399 aa) revealed an aa sequence identity (96.2%) to Monilinia barnavirus J from *M. fructigena* and ~88% to Botrytis cinerea fusarivirus 6 and Sclerotinia sclerotiorum fusarivirus 2. The ORF2-encoded protein (434 aa) revealed aa identity of 93.8% to Monilinia barnavirus J and 70% identity to Sclerotinia sclerotiorum fusarivirus 2. It showed SMC (COG1196) and Spc7 (PF08317) domains.

Phylogenetic analysis yielded two groups of fusariviruses, in agreement with Gilbert et al. [44]. MfrcFV1 and MfrcFV2 were included in Group 2, including other fusariviruses previously detected in *M. fructigena* as well as in other ascomycetes. MfrcFV3 was clustered in a clade of Group 1, including fusariviruses detected in Ascomycetes, and well distinguished by the other clade of the same Group, including fusariviruses from Basidiomycetes (Figure 4).

#### 3.2.5. *Barnaviridae*

A 4993 nt sequence showed similarity to barnaviruses and was provisionally named Monilinia fructicola barnavirus 1 (MfrcBV1). It included three ORFs, ORF1 codes for a peptidase (861 aa), ORF2 for a RdRp (691 aa), and ORF3 for a protein of unknown function (238 aa), with an overlapping of 991 bp and a -1 frameshifting between ORF1 and ORF2. ORF1 contained two conserved domains, a trypsin-like peptidase (PF13365) and a codanin-1 C-terminus (PF15296); ORF2 contained RdRp domains (PF02123 and PF00680), whereas no domains or motifs were detected in the ORF3. The three ORF-encoded proteins showed sequence identities from 94.5% to 99.6% to Monilinia barnavirus A identified in the transcriptome of *M. fructigena* [55] (Table 4).

Phylogenetic analysis of the RdRp clustered MfrcBV1 in a clade with Monilinia barnavirus A, Sclerotinia sclerotiorum barnavirus 1, and *Riboviria* spp. from soil, different from the clade including barnaviruses from basidiomycetes, the type-species Mushroom bacilliform virus LF-1, Rhizoctonia solani barnavirus 1, and Tulasnella barnavirus 1 (Figure 5).

#### 3.2.6. *Benyviridae*

A 5680 nt sequence showed significant similarity to members of the family *Benyviridae* and was provisionally named Monilinia fructicola beny-like virus 1 (MfrcBeLV1). MfrcBeLV1 contained two ORFs (Table 4). The ORF1 encoded a replication-associated protein (1623 aa, 182.96 kDa), with three conserved domains, a Superfamily 1 RNA helicase (PF01443), an RdRp (PF00978), and an Alphavirus-like methyltransferase PROSITE profile (PS51743). The protein showed 47.6% and 40.2% aa sequence identities to Lentinula edodes beny-like virus 1 and Agaricus bisporus virus 13, respectively [7,68]. The ORF2 codes for a hypothetical protein (214 aa; 23.55 kDa) and no conserved domains were detected.

Phylogenetic analysis of the ORF1-encoded protein clustered MfrcBeLV1 in a clade including other mycoviruses and well discriminated by the other two clades, one including viruses from insects and the other from plants (Figure 6).

#### 3.2.7. *Parvoviridae*

Two short sequences of 398 and 329 nt showing similarity to members of the family *Parvoviridae* were identified. Viruses in this family have linear, ssDNA genomes of 4–6 kb and typically infect vertebrates, including humans, and invertebrates. Due to the unexpected association of a parvo-like virus with a fungal host, we confirmed the finding through PCR. Total DNA from single isolates of the pool 4 was analysed and the Serbian Mfrc407 isolate (Table 1) showed to carry the virus. Hence, the region between the two contigs was reconstructed through Sanger sequencing. The new virus was provisionally named Monilinia fructicola parvo-like virus 1 (MfrcPV1). The partial genomic sequence (1617 nt; GenBank accession number ON038373) included incomplete ORFs coding for a nonstructural (NS) protein and a capsid virus protein (VP), which represent the two major gene cassettes in *Parvoviridae* members [69]. The NS protein showed identity of 67.7% to Human CSF-associated densovirus; the VP protein showed identity of 97.5% to a parvo-like virus from the wild bird *Ciconia boyciana* (Table 4). MfrcPV1 is the first member of the *Parviviridae* family detected in a fungal host.

#### 3.2.8. Frequencies of Mycoviral Sequences in *M. fructicola* dsRNAs

The distribution and prevalence of mycoviruses of each family were examined in the pools of *M. fructicola* isolates (Figure 7), each including basically isolates of the same geographical origin (Table 1).

Mitoviruses were largely prevalent with high and quite uniform frequencies in all the pools, representing on average 95.5% of the total row reads, ranging from 73.9% (pool No. 9) to more than 99% (pools No 7, 8, 11, and 12). In detail, MfrcMV2 and MfrcMV1 were abundant in all the pools, followed by MfrcMV10, MfrcMV9, and MfrcMV4, while others (e.g., MfrcMV11 and MfrcMV3) were less represented and their genomes assembled from reads of few pools, such as pool No. 5. The genome sequence of the new putative splipalmivirus MfrcSPV1, mostly reconstructed from reads of the pool No. 9 (South of Italy, Basilicata), was also detected in other pools (i.e., No. 1–4 and 11). Viruses in the family *Botourmiaviridae* were detected at lower frequencies than mitoviruses in all the pools, corresponding to less than 1.1% of the row reads, except for pools No. 9 (22.6%) and No. 10 (6.2%). Overall, MfrcBOV1 and MfrcBOV7 were more frequent than MfrcBOV10, MfrcBOV11, MfrcBOV12, and MfrcBOV8. The latter was present with a low number of row reads only in pool No. 7, including isolates from China. The sequences of the two *Monilinia* hypoviruses were represented by a relatively low number of reads, up to 254 RPKM for MfrcHV1 (pool No. 1) and 315 RPKM for MfrcHV2 (pool No. 6). Similarly, the partial sequence of the putative fusarivirus MfrcFV3 was assembled from reads of pool No. 5 (397 RPKM), whereas the other two fusariviruses were more prevalent in pools No. 6 (17.3 × 10^3^ RPKM) and No. 1–5 (2.5–7.7 × 10^3^ RPKM) for MfrcFV1, and in pool No. 5 (4.9 × 10^3^ RPKM) for MfrcFV2. The new beny-like virus MfrcBLV1 and the barnavirus MfrcBV1 were both more represented in pool No. 6, with RPKM values of 8.4 × 10^2^ and 1.1 × 10^3^, respectively, and at lower rate in the pools No. 1–5. A relatively low number of reads (2.2 × 10^2^ RPKM) mapping on the partially assembled new parvo-like virus MfrcPV1 was exclusively found in pool No. 4, including isolates from Serbia and, in particular, the isolate Mfrc407 hosting the virus.

## 4. Discussion and Conclusions

In this study, a metagenomic-based strategy was used to explore the mycovirome associated with the plant-pathogenic fungus *M. fructicola*, inducing brown rot on stone and pome fruit. The approach combined deep sequencing analysis with Illumina technology, which is highly sensitive in viral detection [7,70,71], and dsRNA-enrichment in samples, to maximize the number of short reads derived from viruses [72]. dsRNA analysis indeed permits the detection of viruses with dsRNA as well as ssRNA or DNA genomes due to generation of transient dsRNA [73,74].

A worldwide collection of 58 *M. fructicola* isolates from several stone fruit (cherry, sour cherry, peach, flat peach, nectarine, and plum) and nine countries, encompassing four continents, were analysed aiming at obtaining the most comprehensive overview on viruses associated with the pathogen. The isolates were grouped in 12 pools basically on their geographical origin, as successfully performed by Ruiz-Padilla et al. [14]. 

Preliminarily, dsRNA extracts from single pools were analysed through gel electrophoresis and diverse patterns with variable numbers of bands were observed, as previously reported for *M. fructicola* [52] and the related species *B. cinerea* and *S. sclerotiorum* [14,45,75].

Illumina sequencing allowed to get a total of 100 Gb, which included 14 Gb (70 million of PE 2 × 150 bp short reads) representing the virome of the analysed isolates. Bioinformatic analysis allowed to reconstruct nearly full-length genomic sequences of 33 viruses. On the ground of their genomic structures, conserved domains, sequence homologies with known viruses, and phylogenetic analysis based on hallmarks in replication associated proteins (RdRp or polyprotein), these could be tentatively classified into three families (*Mitoviridae*, the proposed family *Splipalmiviridae*, and *Botourmiaviridae*) in the phylum *Lenarviricota*, three (*Hypoviridae*, the proposed family *Fusariviridae*, and *Barnaviridae*) in the phylum *Pisuviricota*, and one (*Benyviridae*) in the phylum *Kitrinoviricota*, which were the three phyla characterised by +ssRNA genomes (kingdom *Orthornavirae*) in the realm *Riboviria*. Just one virus had an ssDNA genome and was tentatively classified in the family *Parvoviridae*, phylum *Cossaviricota* of the realm *Monodnaviria*. To our knowledge, this is the first comprehensive analysis of viral diversity in *M. fructicola*. 

Hence, most of the viruses identified in *M. fructicola* had a +ssRNA genome, thus belonging to *Riboviria*, the realm that encloses the largest and most diverse group of eukaryote-infecting viruses, including most plant viruses and many important human and animal pathogens [76]. The prevalence of +ssRNA in *M. fructicola* agrees with previous reports on the diversity and predominance of +ssRNA viruses over -ssRNA and dsRNA viruses in several fungal species [7,14,15,45,68,75]. This might be due to greater infection, replication, and transmission efficiency of these types of viruses in their hosts [77]. Nevertheless, it should be considered that this could be a result of the experimental approaches adopted in fungal growth, nucleic acid extraction, and sequencing procedures that might have favoured the selection of +ssRNA viruses. These may indeed have secondary and tertiary structures, increasing the cellulose binding and protection from nuclease activity [7].

Mycoviruses in the phylum *Lenarviricota* were prevalent in *M. fructicola* isolates, in agreement with previous findings in other fungal plant pathogens [14,15]. These included 12 viruses belonging to the family *Mitoviridae* and 12 viruses belonging to the family *Botourmiaviridae*. Members of the two families are naked +ssRNA viruses with very simple genomes of 2.2 to 4.4 kb encoding a single RdRp [5,78]. 

Viruses of the family *Mitoviridae* occur in fungi and some lead to latent infections while others, such as Botrytis cinerea mitovirus 1 and Sclerotinia sclerotiorum mitovirus 2/KL-1, have obvious phenotypic effects in the fungal hosts, causing swollen and malformed mitochondria and reduced fungal growth and virulence [79,80]. Although mitoviruses have never been reported in *Monilinia* species, we found that these are prevalent components of the virome in *M. fructicola* since they were detected at high frequency in all the analysed samples. This finding agrees with reports for other fungi [14,45,75,81]. Mitoviruses are presumably unaffected by antiviral RNA silencing in the host due to their mitochondrial localization and this might contribute to their widespread occurrence [82]. The 12 mitoviruses detected in *M. fructicola* fall into five distinct clades in the family *Mitoviridae* and this shows their broad genetic diversity. 

Mitoviruses might play an important role in pathosystems other than inducing fungal hypovirulence. Indeed, there are reports of their horizontal transfer from fungi to plants as, for instance, in the case of Botrytis cinerea mitovirus 10 (BcMV10) transmitted from *Botrytis cinerea* to cucumber plants [83]. Moreover, integration events of cDNA of fragments or complete genomes of fungal mitoviruses, likely through a transposon-encoded reverse transcriptase, seems to occur in the mitochondrial DNA of vascular plants (nonretroviral endogenized RNA virus elements; NERVEs) [84]. The outcomes of these processes remain to be investigated. The issue is still debated since bioinformatic analysis revealed the existence of mitoviruses in plants so that a subdivision of the genus *Mitovirus* has been proposed to group viruses from both plants and fungi [85]. 

Monilinia fructicola splipalmivirus 1 (MfrcSPV1) along with the three more strictly related Erysiphe necator associated narnavirus 13, Botrytis cinerea binarnavirus 2, and Plasmopara viticola lesion associated narnavirus 4 were grouped phylogenetically in a single cluster much closer to *Mitoviridae* than *Narnaviridae*. This finding supports the proposed institution of the new family *Splipalmiviridae*, including viruses with segmented genomes coding for RdRp conserved motifs on separated segments [63], and highlights the uncertainty in the discrimination of these taxa.

Mycoviruses in *Botourmiaviridae* are closely related to plant ourmiaviruses and have a wide range of hosts, revealing rapid evolution and high diversity [86]. We found a great diversity in the 12 botourmiaviruses detected in *M. fructicola* that were indeed distributed among the genera *Botoulivirus*, *Magoulivirus*, *Scleroulivirus* [64], and the recently proposed genus *Penoulivirus* [14]. Viruses in this family do not appear to affect fungal hosts although there are few exceptions, such as Fusarium oxysporium ourmia-like virus 1 causing hypovirulence in *Fusarium oxysporum* f.sp. *mormordicae* [87]. ICTV established criteria for demarcation of species and genera in the *Botourmiaviridae* based on RdRp sequence identity lower than 90% and 70%, respectively [64]. Therefore, most detected viruses, except for Monilinia fructicola botourmiaviruses 5, should be new species, and the five Monilinia fructicola botourmiaviruses 6, 7, 8, 11, and 12, should be members of five new distinct genera. These findings suggest the need of a taxonomic revision of the family *Botourmiaviridae*, also taking into consideration that many new viruses have been recently added (e.g., [87,88,89]).

We found two hypoviruses in *M. fructicola*, Monilinia fructicola hypovirus 1 and 2 (MfrcHV1 and MfrcHV2). The phylogenetic analysis divided the family *Hypoviridae* into three major clades, corresponding to the genera *Alphahypovirus* and *Betahypovirus*, with the type species Cryphonectria hypovirus 1 and 2 and Cryphonectria hypovirus 3 and 4, respectively, and the proposed new genus *Gammahypovirus* [80,90]. MrfcHV1 showed 97% sequence identity to Botrytis cinerea hypovirus 1, a member of the genus *Betahypovirus*, inhibiting formation of infection cushions and causing hypovirulence in *B. cinerea* [67,91]. MfrcHV2 showed 85% sequence identity to Sclerotinia sclerotiorum hypovirus 2, inducing morphological and structural alterations and hypovirulence in *S. sclerotiorum,* and should be a member of the genus *Gammahypovirus* [78,90], along with Sclerotium rolfsii hypovirus 1, associated to hypovirulence [24], and Botrytis cinerea hypovirus 5 [14]. Therefore, the results here presented support the proposal of establishing the third genus *Gammahypovirus* in the family.

The three viruses Monilinia fructicola fusarivirus 1, 2, and 3 (MfrcFV1, 2 and 3) were related to the family *Fusariviridae*, proposed but not yet officially approved by ICTV, close to *Hypoviridae*. Phylogenetic analysis divided the family into two major clades, Group 1 and Group 2. This agrees with Gilbert et al. [44] and Picarelli et al. [81], who recognized the two groups due to different genome organization, with Group 1 having the functional domains SMC and Spc7 in the ORF2, as opposed to Group 2 that does not contain any predicted domains. MfrcFV3 should be a member of Group 1, having Fusarium graminearium dsRNA mycovirus-1 as a reference, while MfrcFV1 and MfrcFV2 should be members of Group 2.

*Barnaviridae* typically includes mycoviruses from the Basidiomycetes and has the Mushroom bacilliform virus LF-1/AUS as the type species. *Monilinia fructicola* barnavirus 1 (MfrcBV1) along with the almost identical Monilinia barnavirus A found in a *M. fructigena* transcriptome [55], and Sclerotinia sclerotiorum barnavirus 1 [92], are so far the only members associated with the Ascomycetes. 

There is only the genus *Benyvirus* in the family *Benyviridae*, having four to five +ssRNAs of 6.7, 4.6, 1.8, 1.4, and 1.3 kb [93] and including viruses from plants, insects, and fungi. These were separated in three well-defined clades by phylogenetic analysis. In *M. fructicola*, we could detect, like in Rhizoctonia solani beny-like virus 1 [45], only the RNA1 segment encoding the RdRp protein of Monilinia fructicola benyvirus 1 (MfrcBenV1). This is the first *Bernyvirus* identified in the Ascomycetes and should be a new viral species, showing an RdRp sequence identity lower than 48% with the closest Lentinula edodes ssRNA mycovirus. 

There are only few reports of ssDNA viruses associated with fungi. The few examples are the viruses belonging to the family *Genomoviridae* detected in *S. sclerotiorum* [13], *Fusarium graminearum* [94] and *B. cinerea* [14,20]. In contrast, we found two partial sequences of 329 and 398 nt of ssDNAs showing similarity to members of the *Parvoviridae* family. This family includes ssDNA viruses with a genome sized 4–6 kb and is divided into three subfamilies: *Parvovirinae*, *Densovirinae*, and the recently approved *Hamaparvovirinae* [95]. The host spectrum of these viruses is very broad and includes humans, mammals, fishes, birds, tunicates, arthropods, and flatworms. Some viruses are pathogenic for insects and are used as biocontrol agents [96]. This was a very unexpected result; therefore, the *M. fructicola* isolate hosting the virus Mfrc407 from Nectarine in Serbia was individuated among the pooled isolates. The sequence between the two original fragments was reconstructed by PCR and Sanger sequencing, obtaining a still partial sequence of 1617 nt. It contained two partial ORFs: ORF1 encoding a non-structural replication associated protein and ORF2 encoding the coat protein. ORF1 showed a sequence identity as high as 67% with the Human CFS-associated densovirus and ORF2 a sequence identity of 96.8% to the Ciconia boyciana parvovirus. Then, the virus was tentatively named Monilinia fructicola parvo-like virus 1 (MfrcPLV1), which is the first virus in the *Parvoviridae* family detected in a fungal host. The genome of the virus is incomplete and its characterization and possible role in fungal biology are worthwhile of further research. 

In conclusion, this study provided new insights into the mycovirome diversity of *M. fructicola*, an important pathogen inducing brown rot of stone and pome fruit, thus enhancing the already broad collection of mycoviruses in plant pathogenic fungi. Although many mycoviruses are cryptic and do not alter the phenotypic traits of their fungal hosts, these may be a source of diversity and play important roles in their population biology. The availability of near full-length viral sequences associated with *M. fructicola* will provide crucial tools for investigations into the complex virus–host and virus–virus interactions and improve the background knowledge to explore the potential use of mycoviruses as biocontrol agents, which is certainly worthwhile of further research to obtain to sustainable and environmentally friendly control strategies against this economically important pathogen.

## Figures and Tables

**Figure 1 jof-08-00481-f001:**
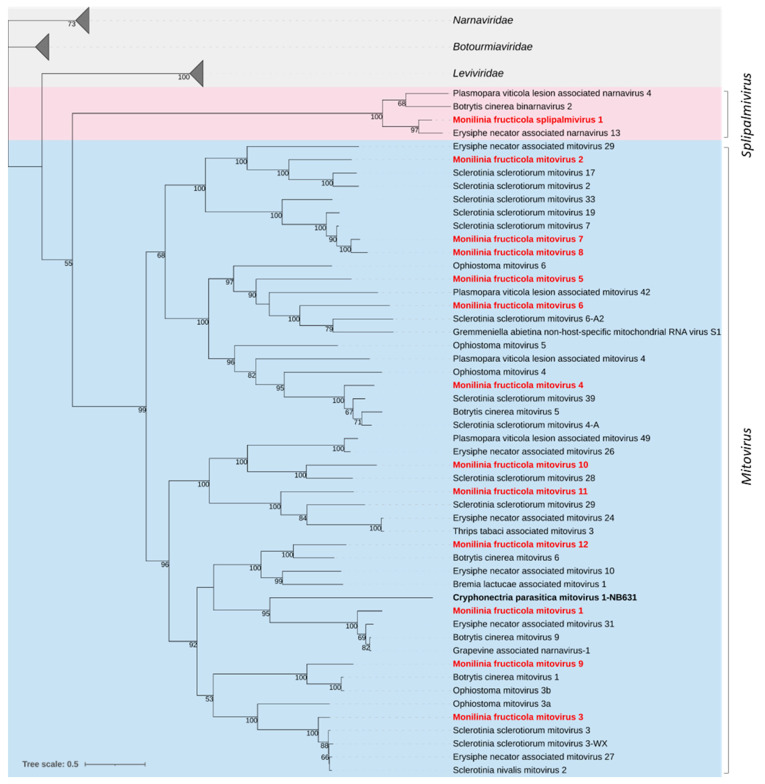
Maximum likelihood phylogenetic tree based on multiple amino acid sequence alignment of the RdRp proteins of viruses in the family *Mitoviridae* (cyan) and the proposed family *Splipalmiviridae* (pink; [14]), and their relationships with the closest families *Leviviridae*, *Narnaviridae*, and *Botourmiaviridae* in the phylum *Lenarviricota*. Mycoviruses found in this work are in bold red. The numbers on nodes, shown for values greater than 50%, are the results of 1000 bootstrap analyses. The scale bar represents a genetic distance of 0.5 amino acid substitutions per site. The list of accession numbers of the analysed sequences is in Appendix A.

**Figure 2 jof-08-00481-f002:**
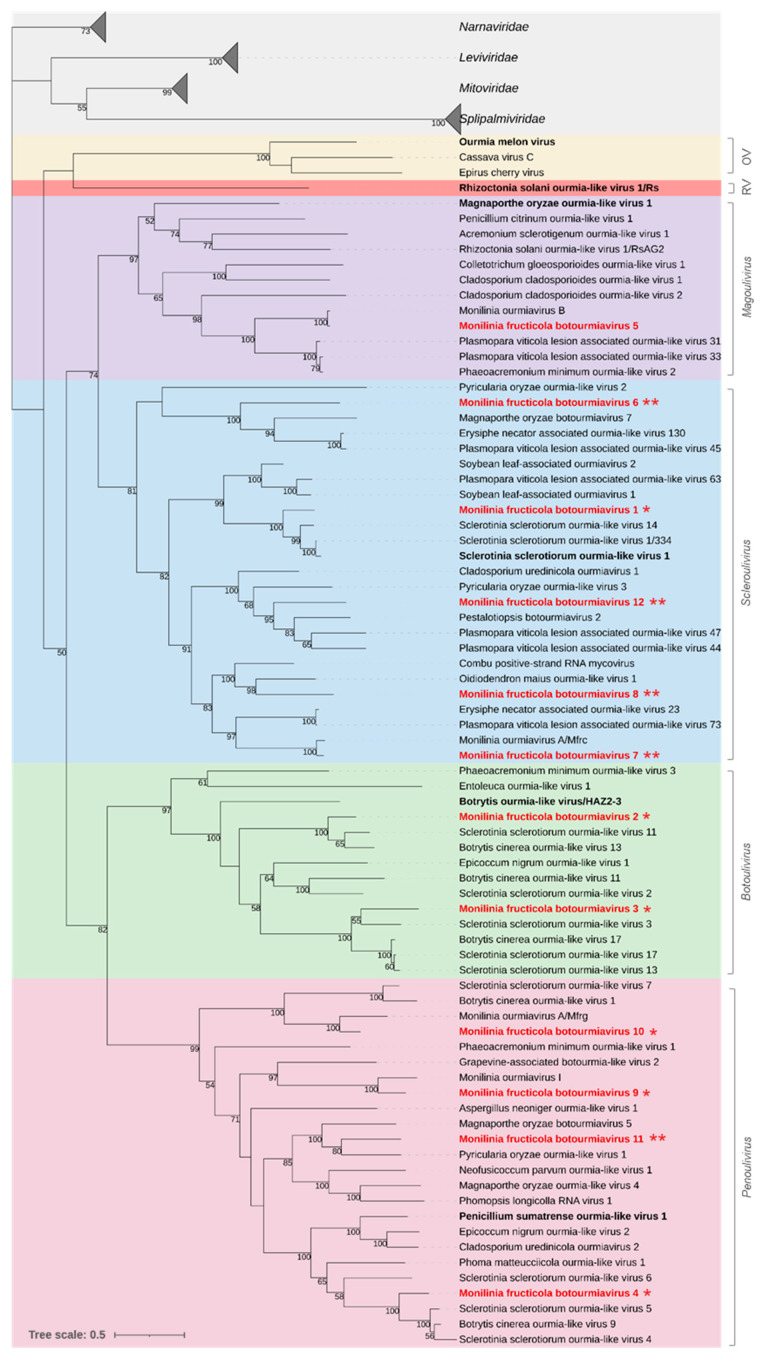
Maximum likelihood phylogenetic tree based on multiple amino acid sequence alignment of the RdRp proteins of viruses in the family *Botourmiaviridae* and its relationships with the closest families *Narnaviridae*, *Leviviridae*, *Splipalmiviridae*, and *Mitoviridae* in the phylum *Lenarviricota.* The genera in the *Botourmiaviridae* family are represented with different coloured boxes: *Ourmiavirus* (OV, yellow), *Rhizoulivirus* (RV, red), *Magoulivirus* (purple), *Scleroulivirus* (cyan), *Botoulivirus* (green), and *Penoulivirus* (pink). The type species is in bold black. Mycoviruses found in this work are in bold red. Single or double asterisks indicate new proposed species and genera, respectively. The numbers on nodes, shown for values greater than 50%, are the results of 1000 bootstrap analyses. The scale bar represents a genetic distance of 0.5 amino acid substitutions per site. The list of accession numbers of the analysed sequences is in the Appendix A.

**Figure 3 jof-08-00481-f003:**
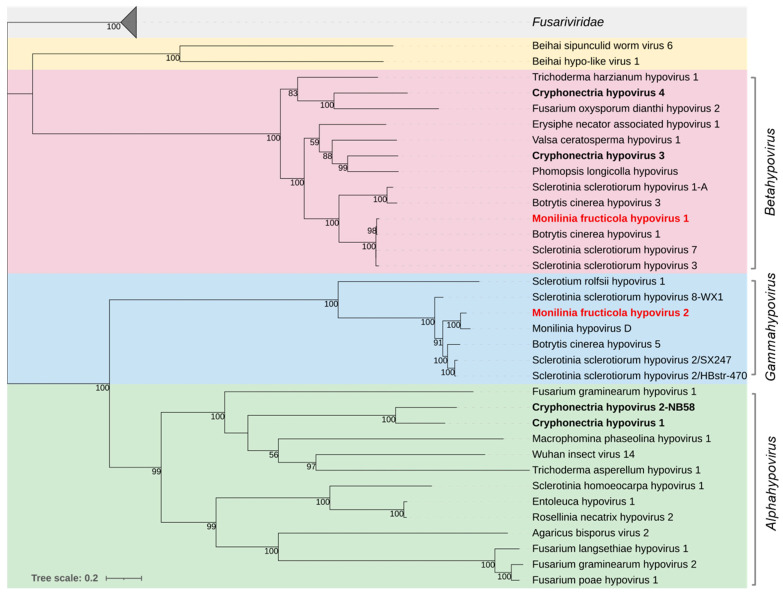
Maximum likelihood phylogenetic tree based on multiple amino acid sequence alignment of the RdRp proteins of viruses in the family *Hypoviridae* and its relationship with the closely related proposed family *Fusariviridae*. The three genera in the *Hypoviridae* family are represented with different coloured boxes: *Alphahypovirus* (green), *Betahypovirus* (red), and the proposed genus *Gammahypovirus* (cyan). The Beihai hypo-like virus and Beihei sipunculid worm virus 6 (yellow) are proposed as members of an additional genus [65]. The type species are in bold black. Mycoviruses found in this work are in bold red. The numbers on nodes, shown for values greater than 50%, are the results of 1000 bootstrap analyses. The scale bar represents a genetic distance of 0.2 amino acid substitutions per site. The list of accession numbers of the analysed sequences is in Appendix A.

**Figure 4 jof-08-00481-f004:**
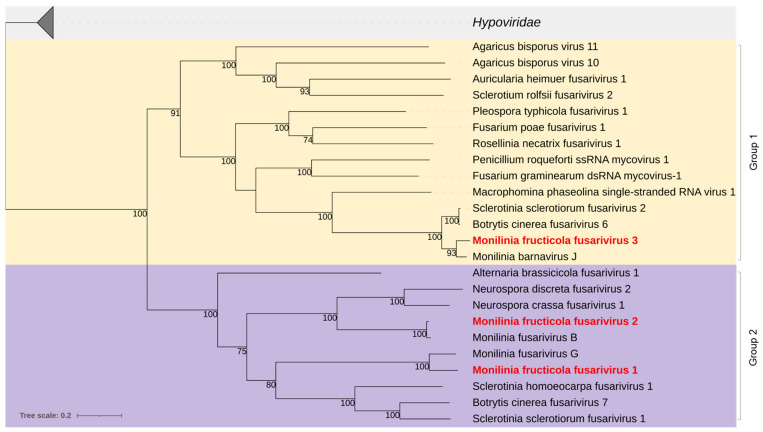
Maximum likelihood phylogenetic tree based on multiple amino acid sequence alignment of the RdRp proteins of viruses in the proposed *Fusariviridae* family and its relationship with the closely related family *Hypoviridae*. Coloured boxes indicate Group 1 (yellow) and Group 2 (purple) of the fusariviruses [44]. Mycoviruses found in this work are in bold red. The numbers on nodes, shown for values greater than 50%, are the results of 1000 bootstrap analyses. The scale bar represents a genetic distance of 0.2 amino acid substitutions per site. The list of accession numbers of the analysed sequences is in Appendix A.

**Figure 5 jof-08-00481-f005:**
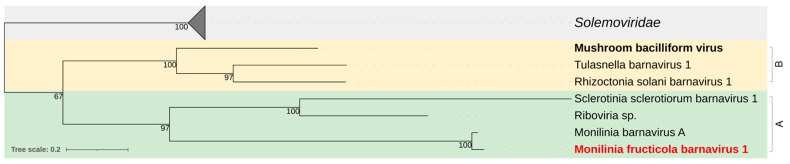
Maximum likelihood phylogenetic tree based on multiple amino acid sequence alignment of the replicase protein of Monilinia fructicola barnavirus 1 and viruses of the genus *Barnavirus* (*Barnaviridae* family) reported in Basidiomycetes (B; yellow) and Ascomycetes (A; green), and the closely related members of the *Solemoviridae* family (grey). The type species is in bold black. The mycovirus found in this work is in bold red. The number on nodes, shown for values greater than 50%, are the results of 1000 bootstrap analyses. The scale bar represents a genetic distance of 0.2 amino acid substitutions per site. The list of accession numbers of the analysed sequences is in Appendix A.

**Figure 6 jof-08-00481-f006:**
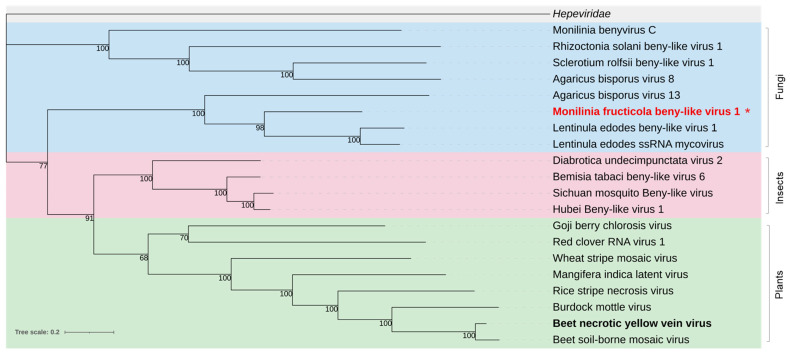
Maximum likelihood phylogenetic tree based on multiple amino acid sequence alignment of the nonstructural replication-associated protein of Monilinia fructicola beny-like virus 1 and beny-like viruses (*Benyviridae* family) from various hosts, including fungi (blue), insects (pink), and plants (green). The *Hepeviridae* family (grey) was used as outgroup. The type species is in bold black. The mycovirus found in this work is in bold red. The asterisk indicates the newly proposed species. The number on the nodes, shown for values greater than 50%, are the results of 1000 bootstrap analyses. The scale bar represents a genetic distance of 0.2 amino acid substitutions per site. The list of accession numbers of the analysed sequences is in Appendix A.

**Figure 7 jof-08-00481-f007:**
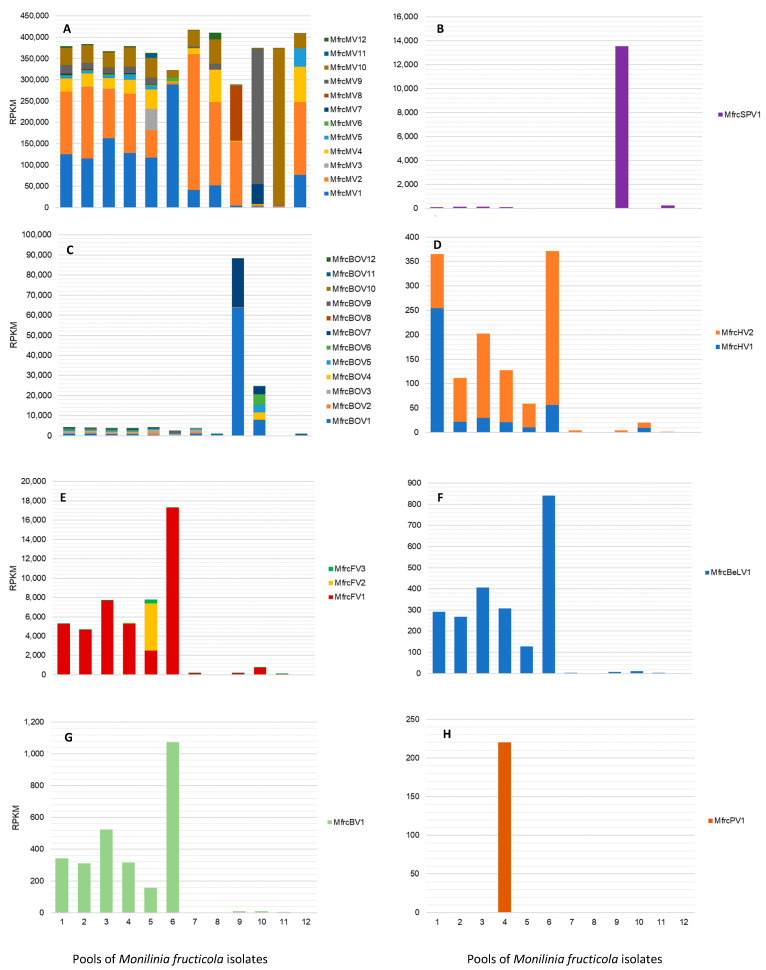
Detection rate of viral families and single viruses in 12 pools of *Monilinia fructicola* isolates in RPKM (reads normalized per kilobases of viral genome length and millions of total reads mapped on the virome). (**A**) *Mitoviriridae*; (**B**) *Splipalmiviridae*; (**C**) *Botourmiaviridae*; (**D**) *Hypoviridae*; (**E**) *Fusariviridae*; (**F**) *Benyviridae*; (**G**) *Barnaviridae*; (**H**) *Parvoviridae*.

**Table 1 jof-08-00481-t001:** Pools of *M. fructicola* strains used in the mycovirome characterization.

Pool No.	Isolate Code	Original Code	Location	Host Plant	Year	Provided by
1	Mfrc426	LSV M 177	France (Provence-Alpes-Cotes d’Azur)	Apricot	2010	C. Guinet, France
	Mfrc427	LSV M 178	France (Provence-Alpes-Cotes d’Azur)	Cherry	2008	
	Mfrc428	LSV M 1061	France (Provence-Alpes-Cotes d’Azur)	Peach	2010	
	Mfrc429	LSV M 1062	France (Provence-Alpes-Cotes d’Azur)	Peach	2010	
	Mfrc430	LSV M 1063	France (Provence-Alpes-Cotes d’Azur)	Peach	2010	
2	Mfrc362	F5	Greece (Imathia, Central Macedonia)	Peach	Unknown	G. Karaoglanidis, Greece
	Mfrc363	F15	Greece (Imathia, Central Macedonia)	Peach	Unknown	
	Mfrc364	A6	Greece (Pella, Central Macedonia)	Peach	Unknown	
	Mfrc365	A7	Greece (Pella, Central Macedonia)	Peach	Unknown	
3	Mfrc416	2709	New Zealand (Central Otago)	Apricot	1969	B. Weir, New Zealand
	Mfrc418	7640	New Zealand (Royal Oak—Auckland)	Peach	Unknown	
	Mfrc419	7641	New Zealand (Whenuapai—Auckland)	Peach	1979	
	Mfrc420	7642	New Zealand (Hastings—Hawke’s Bay)	Peach	1979	
	Mfrc424	20117	New Zealand (Christchurch, Mid Canterbury)	Plum	2014	
4	Mfrc407	I/1 NPUD	Serbia (Udovice, Smederevo)	Nectarine	2012	J. Hrustic, Serbia
	Mfrc408	I/1 SPOS	Serbia (Osečina, Kolubara)	Plum	2013	
	Mfrc410	I/2 TPST	Serbia (Smederevo)	Cherry	2014	
	Mfrc412	8 TP/28	Serbia (Požarevac, Braničevo)	Cherry	2015	
	Mfrc414	27 BP/12	Serbia (Golobok, Smederevska Palanka)	Peach	2016	
5	Mfrc354	M4.C7.14	USA (Clemson, South Carolina)	Peach	2016	G. Schnabel, USA
	Mfrc355	M2.C1.1	USA (Clemson, South Carolina)	Peach	2016	
	Mfrc356	Z3.C10.5	USA (Sandy Springs, South Carolina)	Peach	2016	
	Mfrc357	Z4.C3.3	USA (Sandy Springs, South Carolina)	Peach	2016	
	Mfrc358	Z1.C6.4	USA (Sandy Springs, South Carolina)	Peach	2016	
6	Mfrc391	21C	Spain (Lleida, Alcarràs)	Unknown	2013	A. De Cal, Spain
	Mfrc393	23C	Spain (Lleida, Alcarràs)	Unknown	2013	
	Mfrc394	34C	Spain (Lleida, Albesa)	Unknown	2009	
	Mfrc396	37C	Spain (Lleida, Alfarràs)	Unknown	2009	
	Mfrc397	43C	Spain	Unknown	2011	
7	Mfrc435	cfcc 80267	China (Linyi City, Shandong Province)	Peach	2005	L.-Y. Guo, China
	Mfrc436	cfcc 80268	China (Yantai City, Shandong Province)	Peach	2005	
	Mfrc437	cfcc 80248	China (Chaoyang, Pechino, Beijing)	Peach	2005	
	Mfrc438	cfcc 80519	China (Fangshan, Pechino, Beijing)	Plum	2005	
8	Mfrc106	C1	Italy (Bisceglie, Apulia)	Cherry	2014	Our fungal collection, Italy
	Mfrc123	C18	Italy (Bisceglie, Apulia)	Cherry	2014	
	Mfrc301	M16	Italy (Gioia del Colle, Apulia)	Cherry	2014	
	Mfrc322	O17	Italy (Gioia del Colle, Apulia)	Cherry	2014	
	Mfrc376	5	Italy (Cerignola, Apulia)	Peach	2015	
9	Mfrc148	E3	Italy (Tursi, Basilicata)	Plum	2014	Our fungal collection, Italy
	Mfrc150	E5	Italy (Tursi, Basilicata)	Plum	2014	
	Mfrc247	L2	Italy (Policoro, Basilicata)	Flat peach	2014	
	Mfrc261	L16	Italy (Policoro, Basilicata)	Flat peach	2014	
	Mfrc373	2	Italy (Loconia, Basilicata)	Peach	2014	
10	Mfrc350	VA bf P12m	USA (Oak Grove, Virginia)	Plum	2012	G. Schnabel, USA
	Mfrc352	BMPC 10	USA (Byron, Georgia)	Peach	2006	
	Mfrc401	cc866	USA	Plum	1995	G.C.M. Van Leeuwen, The Netherlands
	Mfrc402	cc867	USA	Plum	1995	
	Mfrc534	F534	USA	Apricot	Unknown	F. Nigro, Italy
11	Mfrc331	03-K47	USA (Parlier, California)	Peach	2002	T. Michailides, USA
	Mfrc347	NY 9C	USA (Geneva, New York)	Cherry	2007	G. Schnabel, USA
	Mfrc351	VA bf P16s	USA (Oak Grove, Virginia)	Plum	2012	
	Mfrc404	dar27031	Australia	Peach	1995	G.C.M. Van Leeuwen, The Netherlands
	Mfrc405	NZ 2.89	New Zealand	Plum	1995	
12	Mfrc77	A11	Italy (Caserta, Campania)	Cherry	2014	Our fungal collection, Italy
	Mfrc78	A12	Italy (Caserta, Campania)	Cherry	2014	
	Mfrc395	35C	Spain (Lleida, Albesa)	Unknown	2009	A. De Cal, Spain
	Mfrc399	45C	Spain	Unknown	2011	
	Mfrc415	3 VP/1L	Serbia (Šabac, Mačva)	Sour cherry	2016	J. Hrustic, Serbia

**Table 2 jof-08-00481-t002:** Summary statistics of the sequencing data.

Pool	Total No. of Paired End (PE) Reads	PE Reads Filtered for Quality (QS ≥ 30)			GC%
		Total No.	Nonredundant		
			No.	%	
1	21,633,810	21,483,924	15,579,294	72.5	29
2	20,710,678	20,562,344	17,423,394	84.7	34
3	22,653,808	22,481,652	12,863,610	83.9	34
4	31,855,794	31,616,498	26,697,938	84.4	29
5	21,628,744	21,482,266	16,170,072	75.3	30
6	106,258,396	105,125,504	26,667,956	25.4	40
7	85,651,472	84,768,112	21,327,038	25.2	41
8	87,698,960	86,789,590	21,053,874	24.2	44
9	80,371,552	79,533,392	20,619,356	25.9	47
10	66,807,410	66,139,758	24,827,978	37.5	43
11	62,570,222	61,947,854	24,285,582	39.2	43
12	72,161,200	71,415,466	15,575,196	21.8	40

**Table 3 jof-08-00481-t003:** Summary of the de novo assembly and BLASTP search of contigs.

Pool	De Novo Assembly			No. of Putative Viral Contigs Selected by BLASTP Analysis	
	Contigs (No.)	Mapped Reads		Total	Nonredundant
		(No.) ^¥^	(%)		
1	48,825	19,461,803	90.59	1527	128
2	34,131	17,854,177	86.83	1661	127
3	53,832	19,461,803	86.57	1612	111
4	49,458	28,213,117	89.24	1773	138
5	61,986	17,595,205	81.91	1510	193
6	43,774	103,533,111	98.49	2058	588
7	46,702	82,089,562	96.84	551	508
8	28,049	84,503,137	97.37	163	106
9	42,092	78,390,075	98.56	553	67
10	43,095	63,088,469	95.39	1725	145
11	29,999	59,945,146	96.77	1622	106
12	31,740	68,867,450	96.43	1308	93

^¥^ Numbers include both reads in pair or broken reads mapped on the assembled contigs.

**Table 4 jof-08-00481-t004:** Putative mycoviruses identified in *Monilinia fructicola*.

Virus	Genome (nt)	Protein (aa)	Similarity with Viral Sequences (BLASTP)			
			Mycovirus	Coverage	E-Value	Identity
*Mitoviridae*						
MfrcMV1	+ssRNA (2658)	RdRp (721)	Botrytis cinerea mitovirus 9	100%	0	81.97%
MfrcMV2	+ssRNA (2298)	RdRp (683)	Sclerotinia sclerotiorum mitovirus 2	97%	0	63.34%
MfrcMV3	+ssRNA (2657)	RdRp (712)	Sclerotinia sclerotiorum mitovirus 3-WX	100%	0	92.13%
MfrcMV4	+ssRNA (2441)	RdRp (731)	Sclerotinia sclerotiorum mitovirus 39	97%	0	84.11%
MfrcMV5	+ssRNA (2430)	RdRp (698)	Ophiostoma mitovirus 6	94%	3 × 10^−162^	42.28%
MfrcMV6	+ssRNA (2341)	RdRp (710)	Sclerotinia sclerotiorum mitovirus 46	98%	0	69.86%
MfrcMV7	+ssRNA (2718)	RdRp (689)	Sclerotinia sclerotiorum mitovirus 7	100%	0	99.13%
MfrcMV8	+ssRNA (2758)	RdRp (689)	Sclerotinia sclerotiorum mitovirus 19	99%	0	90.96%
MfrcMV9	+ssRNA (2438)	RdRp (738)	Botrytis cinerea mitovirus 1	100%	0	79.76%
MfrcMV10	+ssRNA (2670)	RdRp (794)	Sclerotinia sclerotiorum mitovirus 28	97%	0	70.82%
MfrcMV11	+ssRNA (2816)	RdRp (866)	Sclerotinia sclerotiorum mitovirus 29	99%	0	64.60%
MfrcMV12	+ssRNA (2401)	RdRp (708)	Botrytis cinerea mitovirus 6	99%	0	78.25%
*Splipalmiviridae*						
MfrcSPV1	+ssRNA (2466)	RdRp (785)	Erysiphe necator associated narnavirus 13	100%	0	77.23%
	+ssRNA (2123) ^1^	Hypothetical protein (697)	Botrytis cinerea binarnavirus 2	93%	0	47.87%
*Botourmiaviridae*						
MfrcBOV1	+ssRNA (2830)	RdRp (684)	Sclerotinia sclerotiorum ourmia-like virus 1	99%	0	72.70%
MfrcBOV2	+ssRNA (2917)	RdRp (707)	Botrytis cinerea ourmia-like virus 13	100%	0	73.38%
MfrcBOV3	+ssRNA (2520)	RdRp (647)	Sclerotinia sclerotiorum ourmia-like virus 13	99%	0	70.33%
MfrcBOV4	+ssRNA (2617)	RdRp (730)	Botrytis cinerea ourmia-like virus 9	91%	0	78.85%
MfrcBOV5	+ssRNA (2792)	RdRp (656)	Monilinia ourmiavirus B	100%	0	99.70%
MfrcBOV6	+ssRNA (3004)	RdRp (655)	Erysiphe necator associated ourmia-like virus 130	93%	2 × 10^−172^	46.96%
MfrcBOV7	+ssRNA (2382)	RdRp (645)	Plasmopara viticola associated ourmia-like virus 73	94%	2 × 10^−170^	47.66%
MfrcBOV8	+ssRNA (2695)	RdRp (668)	Oidiodendron maius ourmia-like virus 1	95%	0	54.25%
MfrcBOV9	+ssRNA (2638)	RdRp (780)	Monilinia ourmiavirus I	69%	0	86.64%
MfrcBOV10	+ssRNA (2774)	RdRp (736)	Monilinia ourmiavirus A	99%	0	85.91%
MfrcBOV11	+ssRNA (2438)	RdRp (667)	Pyricularia oryzae ourmia-like virus 1	97%	0	54.71%
MfrcBOV12	+ssRNA (2544)	RdRp (654)	Plasmopara viticola associated ourmia-like virus 47	97%	0	50.54%
*Hypoviridae*						
MfrcHV1	+ssRNA (9338)	Polyprotein (2918)	Sclerotinia sclerotiorum hypovirus 7	100%	0	97.74%
MfrcHV2	+ssRNA (15,037)	Polyprotein (4784)	Monilinia hypovirus D	99%	0	93.24%
*Fusariviridae*						
MfrcFV1	+ssRNA (7425)	RdRp (1661)	Monilinia fusarivirus G	99%	0	81.24%
		Protein ShFV1_gp2 (504)	Monilinia fusarivirus G	98%	0	71.54%
MfrcFV2	+ssRNA (7330)	RdRp (1667)	Monilinia fusarivirus B	100%	0	97.36%
		Protein BSB06_gp2 (509)	Monilinia fusarivirus B	100%	0	95.48%
MfrcFV3	+ssRNA (5550) ^1^	RdRp (1399)	Monilinia barnavirus J	100%	0	96.21%
		Protein ShFV1_gp2 (434)	Sclerotinia sclerotiorum fusarivirus 2	99%	0	70.00%
*Barnaviridae*						
MfrcBV1	+ssRNA (4993)	Peptidase (861)	Monilinia barnavirus A ORF1	99%	0	94.52%
		RdRp (691)	Monilinia barnavirus A ORF2	100%	0	94.50%
		Putative protein (238)	Monilinia barnavirus A ORF3	100%	4 × 10^−170^	99.58%
*Benyviridae*						
MfrcBeLV1	+ssRNA (5761)	Replication-associated protein (1623)	Lentinula edodes ssRNA mycovirus	100%	0	47.38%
		Hypothetical protein (214)	-	-	-	-
*Parvoviridae*						
MfrcPV1	ssDNA (1664) ^1^	Non-structural protein NS3 (152)	Human CSF-associated densovirus	94%	3 × 10^−64^	67.13%
		Structural protein VP (375)	Ciconia boyciana parvoviridae	100%	0	96.80%

^1^ Partial sequence.

## Data Availability

All raw sequencing reads were stored in the Sequence Read Archive (SRA) database: BioProject accession number PRJNA818196, BioSample accession numbers from SAMN26849489 to SAMN26849500, and SRA runs from SRX14530422 to SRX14530433. Genome sequences of the mycoviruses identified have been deposited in GenBank under accession numbers ON038373 and ON038375—ON038406 (detailed in Appendix A).

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
