# Peer review of "The Mycovirome in a Worldwide Collection of the Brown Rot Fungus Monilinia fructicola"

_jof, 2022, doi:10.3390/jof8050481_

Round 1
Reviewer 1 Report
In this study, a metagenomic approach was applied to obtain a comprehensive characterization of the mycovirome in M. fructicola strains. This study provided new insights on the mycovirome diversity in M. fructicola to some extent. The scientific innovation is limited. It exists some problems to be addressed.
- The obtained +ssRNA viral sequences need to be identified by RT-PCR using the sequencing M. fructicola strains as templates. Therefore, the RT-PCR results need to be provided.
2.The writing is sometimes intricate. In some sections it is not clear. English need to be modified.
- The references in the manuscript should be checked.
Author Response
Reviewer 1: In this study, a metagenomic approach was applied to obtain a comprehensive characterization of the mycovirome in M. fructicola strains. This study provided new insights on the mycovirome diversity in M. fructicola to some extent. The scientific innovation is limited. It exists some problems to be addressed.
- The obtained +ssRNA viral sequences need to be identified by RT-PCR using the sequencing M. fructicola strains as templates. Therefore, the RT-PCR results need to be provided.
Authors: The present work was a survey carried out on a in a worldwide collection of fungal isolates aimed at a comprehensive reconstruction of the mycovirome of M. fructicola using a metagenomic approach. A validation of the de novo assembled sequences by PCR and Sanger sequences was performed only for selected viruses showing low level of sequence identity with known viruses or never reported in fungi, as in the case of the putative ssDNA parvovirus. A deeper characterization of the detected virus, including the obtainment of their full-length sequences is in progress.
- The writing is sometimes intricate. In some sections it is not clear. English need to be modified.
Authors: Thanks to the reviewer for the suggestion. The manuscript has been carefully checked for English.
- The references in the manuscript should be checked.
Authors: The suggestion has been accepted and the references checked.
Reviewer 2 Report
The manuscript entitled "The mycovirome in a worldwide collection of the brown rot fungus Monilinia fructicola" aims to reveal the presence of mycoviruses in a collection of Monilinia fructicola isolates collected from 9 different countries. The authors used the advent of Next generation sequencing and detected a substantial number of contigs with similarities to mycoviruses representing 33 virus-like genomes most of which are novel.This manuscript reports for the first time a DNA virus related to parvoviruses. Overall, the manuscript is well written and the introduction covers the basic information related to the purpose of the study providing some background information on mycoviruses and their importance in fungal virology & fungus-host relationship, the devastating phytopathogen Monilinia fructicola and its mycoviruses, and how NGS techniques have employed in the study of mycoviruses. Materials and methods were written in details. Some further experiments such as obtaining the full-length sequences of the viruses detected in this study would have improved the manuscript. However, since this paper is a survey and sequencing paper, the approaches used are sufficient. Results and discussion are well presented. I personally have an issue with the way some viruses were named. It would have been more appropriate to refer to them as "sequences; i.e. sequence 1, 2, 3, ........." then when this sequence shares high similarity with known viruses of the same host (as per the demarcation criteria), they can be considered isolates of the same species. Here, some viruses that are >99% similar to previously characterized viruses were given new names (i.e. in lines 336, 498, 502, etc). Presenting a gel picture for the dsRNA profile of the pools and correlating that with the sequences detected would have improved the presentation of the results. The DNA profile is even more important for the unusually detected DNA virus found in only 1 isolate.
Author Response
Comments and Suggestions for Authors (Reviewer 2)
The manuscript entitled "The mycovirome in a worldwide collection of the brown rot fungus Monilinia fructicola" aims to reveal the presence of mycoviruses in a collection of Monilinia fructicola isolates collected from 9 different countries. The authors used the advent of Next generation sequencing and detected a substantial number of contigs with similarities to mycoviruses representing 33 virus-like genomes most of which are novel.
This manuscript reports for the first time a DNA virus related to parvoviruses. Overall, the manuscript is well written and the introduction covers the basic information related to the purpose of the study providing some background information on mycoviruses and their importance in fungal virology & fungus-host relationship, the devastating phytopathogen Monilinia fructicola and its mycoviruses, and how NGS techniques have employed in the study of mycoviruses. Materials and methods were written in details. Some further experiments such as obtaining the full-length sequences of the viruses detected in this study would have improved the manuscript. However, since this paper is a survey and sequencing paper, the approaches used are sufficient. Results and discussion are well presented.
Authors: The present work was indeed a survey carried out on a in a worldwide collection of fungal isolates aimed at a comprehensive reconstruction of the mycovirome of M. fructicola using a metagenomic approach. A deeper characterization of the detected virus, including the obtainment of their full-length sequences is in progress.
Reviewer 2: I personally have an issue with the way some viruses were named. It would have been more appropriate to refer to them as "sequences; i.e. sequence 1, 2, 3, ........." then when this sequence shares high similarity with known viruses of the same host (as per the demarcation criteria), they can be considered isolates of the same species. Here, some viruses that are >99% similar to previously characterized viruses were given new names (i.e. in lines 336, 498, 502, etc).
Authors: We agree with the reviewer that viruses showing high similarity with viruses known in the same host should be considered as strains of the same viral species rather than new viruses. The viruses that we designated Monilinia fructicola botourmiavirus 9 and 10 (MfrcBOV9 and 10), Monilinia fructicola hypovirus 2 (MfrcHV2), Monilinia fructicola fusarivirus 1, 2 and 3 (MfrcFV1, 2, and 3), and Monilinia fructicola barnavirus 1 (MfrcBV1) showed high aa sequence identity with viruses identified in Monilinia fructigena (Monilinia ourmiavirus I and A, Monilinia hypovirus D, Monilinia fusarivirus G and B, Monilinia barnavirus J, and Monilinia barnavirus A) but never reported in Monilinia fructicola. The only exception was Monilinia fructicola botourmiavirus 5 which was highly similar with Monilinia ourmiavirus B reported in M. fructicola. However, we prefer to maintain the name ‘Monilinia fructicola botourmiavirus 5’ also for this virus to respect the classification adopted for the other viruses. We would like to point out that all the previously reported viruses were from in silico assembly of our Illumina RNA-Seq data, obtained for transcriptome characterization and comparison among Monilinia species, available in NCBI database. The related paper “Jo, Y.; Choi, H.; Chu, H.; Cho, W.K. Identification of viruses from fungal transcriptomes. BioRxiv 2020, preprint. DOI: 10.1101/2020.02.26.966903” remained a preprint. Nevertheless, the viral sequences are available in NCBI database, and we preferred to include these in our analysis.
Reviewer 2: Presenting a gel picture for the dsRNA profile of the pools and correlating that with the sequences detected would have improved the presentation of the results. The DNA profile is even more important for the unusually detected DNA virus found in only 1 isolate.
Authors: The suggestion was partially accepted. A gel picture showing the dsRNA profiles for some pools of isolates has been added to the manuscript as Supplementary Figure 1 (Lines 235-236). It should be noted that major bands (1-3 per pool) with size ranging from 2.5 to 3.0 kb likely correspond to mitoviruses, largely prevalent in the mycovirome of M. fructicola in term of viral density. Both NGS and PCR analysis suggest that the putative ssDNA parvovirus is present at very low density in the fungal host and is not detectable by agarose gel electrophoresis.
Reviewer 3 Report
In the manuscript entitled "The mycovirome in a worldwide collection of the brown rot fungus Monilinia fructicola", the authors analyzed the mycovirome of the phytopathogenic fungus Monilinia fructicola. They performed a dsRNA-Seq from 58 M.fructicola strains from different global localization and different hosts. They found a total of 33 viruses. One of them is a putative ssDNA virus from the Parvoviridae family that never was founded in fungi.
In general, the article is interesting and innovative, particularly for the research community of Monilinea fructicola. Although there are no very significant findings, it is always interesting to be able to look for alternatives that help control this phytopathogenic fungus. A good knowledge of the viruses present in Monilinia fruticola is essential for looking for mycoviral alternatives for biocontrol.
Although the article is correctly written and methodologically correct, I have some comments that might help to improve the article even more.
Main critiques
It is a shame that, perhaps due to a technical limitation, neither negative-stranded RNA viruses nor double-stranded RNA viruses could be found. They do not find negative or double-stranded RNA viruses. Is there a possible explanation for this? It is possible that all this insolated fungus does not have these types of virus, or it is simply a technical problem.
Regarding the putative ssDNA virus founded:
How do you explain that this virus is found in a dsRNA library?
Is there a possible contamination with ssRNA or dsDNA, which would explain the detection of the ssDNA virus?
In what phase of the ssDNA virus is there a dsRNA structure?
Is there a possibility that the possible ssDNA virus is a fragment that is inserted into the genome of the fungus?
As this finding is somewhat surprising, it would be convenient to ensure that it is really this virus and not some artifact or contamination. Is it possible to sequence or detect the entire genome of the ssDNA virus? For example: Detect the size of the entire genome using a Southern blot experiment using the region determined by PCR as a probe.
This type of virus from the Parvoviridae family has a length between 4 and 6 kb. If the virus is inside M.fruticola, its genome must be full lenght.
In this sentence:
Ln 719 “dsRNA analysis permits indeed detection of viruses with dsRNA as well as ssRNA or DNA genomes due to generation of abundant transient dsRNA intermediates during virus replication” REF [48].
I guess reference 48 is not related to this assertion. Please confirm that this sentence is related to this reference.
A figure to clarify the way in which dsRNA sequencing has been attempted and a diagram of some virus from the Parvoviridae family would be appreciated.
Is there a phenotype associated with the isolates that present this ssDNA virus?
Are there different phenotypes in all isolates used in this article?
The possible origin of this possible virus in the fungus is not explained. Therefore, it might be interesting to discuss the possible origin of this novel virus in fungi.
In other families, it would be good to include at least a diagram of the genome of the most representative viruses found.
About Binarnavirus: Recently a new article by Sato, Y. et al. was published. 'A new tetra-segmented splipalmivirus with divided RdRP domains from Cryphonectria naterciae, a fungus found on chestnut and cork oak trees in Europe'. Virus Res. 307, 198606 (2022).
They reported in C. naterciea a positive-sense (+), single-stranded (ss) RNA viruses with divided RNA-dependent RNA polymerase (RdRP) domains. These viruses are termed splipalmiviruses or polynarnaviruses and I think that will be a better name instead of binarnavirus because CnSpV1 has a tetra-segmented genome. A (+)ssRNA genome (RNA1 to RNA4). The RdRP domain is separately encoded by RNA1 (motifs F, A, and B) and RNA2 (motifs C and D). RNA3 encoded a hypothetical protein shows similarity to the conserved counterpart in some splipalmiviruses. And the RNA4 encoded protein does not show similarity to known proteins. Therefore, did you find RNA2, RNA3, and RNA4 in your dsRNA-seq?
Possibly it would be more correct to change Monilinia fructicola binarnavirus 1 name by another name such as splipalmiviruses or polynarnaviruses. Please check Sato, Y. et al. to confirm which narnavirus is Monilinia fructicola binarnavirus 1 according to its segments.
Minor critiques
I found an error in figure number 7. I guess panel G is for Barnaviridae and panel H is for Parvoviridae.
In all phylogenetic trees the Bootstrap % are not easy to read.
I miss all the accession numbers for new viruses and for viruses used in phylogenetic trees. It would be great to include a new table with all accession lists and all characteristics of the virus annotated (e.g., length, proteins, segment, etc).
Conclusions
Overall, I think the article is interesting in the field and can provide information to the scientific community working on this phytopathogenic fungus. Although the article does not present major findings, there is a comprehensive description of Monilinea fruticola mycovirome. In any case, I think the article could be improved. One of the aspects that should be improved is regarding the finding of the ssDNA virus. I guess it is necessary to confirm that it is really this complete virus inside the fungus and that it is not an artifact, or a sequence inserted in the fungal genome. For such issue, I recommend a major revision to confirm this ssDNA virus is replicating inside Monilinea fruticola cell. I also think it would be interesting to try to connect the presence of the virus with some aspect of the phenotype (e.g., virulence). And finally, it would be great to confirm the multisegmented genome of the narnavirus Monilinia fructicola binarnavirus 1.
Author Response
Reviewer 3: In the manuscript entitled "The mycovirome in a worldwide collection of the brown rot fungus Monilinia fructicola", the authors analyzed the mycovirome of the phytopathogenic fungus Monilinia fructicola. They performed a dsRNA-Seq from 58 M. fructicola strains from different global localization and different hosts. They found a total of 33 viruses. One of them is a putative ssDNA virus from the Parvoviridae family that never was founded in fungi.
In general, the article is interesting and innovative, particularly for the research community of Monilinea fructicola. Although there are no very significant findings, it is always interesting to be able to look for alternatives that help control this phytopathogenic fungus. A good knowledge of the viruses present in Monilinia fruticola is essential for looking for mycoviral alternatives for biocontrol.
Although the article is correctly written and methodologically correct, I have some comments that might help to improve the article even more.
Authors: We are very grateful to the referee for the careful reading of the manuscript and for his comments and detailed suggestions which helped us to improve considerably the manuscript.
Main critiques
Reviewer 3: It is a shame that, perhaps due to a technical limitation, neither negative-stranded RNA viruses nor double-stranded RNA viruses could be found. They do not find negative or double-stranded RNA viruses. Is there a possible explanation for this? It is possible that all this insolated fungus does not have these types of virus, or it is simply a technical problem.
Authors: The reviewer is right that unexpectedly we did not find any dsRNA or -ssRNA viruses in our survey. However, this finding agrees with previous reports on the high prevalence of +ssRNA viruses over -ssRNA and dsRNA viruses in several fungal species, as we discussed at lines 747-750 of the manuscript. This might be due to differences in infection rate and in replication and transmission mechanisms that might be more efficient in +ssRNA viruses as compared to other types of viruses. On the other hand, we cannot exclude ‘technical limitations’ of the experimental strategy adopted as it is stated at lines 843-847). “Nevertheless, it should be considered that this could be a result of the experimental approaches adopted in fungal growth, nucleic acid extraction and sequencing procedures that might have favoured the selection of +ssRNA viruses. These may indeed have secondary and tertiary structures increasing cellulose binding and protection from nuclease activity”.
Reviewer 3: Regarding the putative ssDNA virus founded:
How do you explain that this virus is found in a dsRNA library?
Is there a possible contamination with ssRNA or dsDNA, which would explain the detection of the ssDNA virus?
In what phase of the ssDNA virus is there a dsRNA structure?
Authors: It is very unlikely that the sequences related to the ssDNA virus were derived from contamination with ssRNA or dsDNA since dsRNA extracts were treated with both DNAase I and ribonuclease A (lines 168-169). Moreover, the presence of the virus in DNA from single isolates was confirmed by PCR and Sanger sequencing. dsRNA can be detected in both RNA and DNA virus infections. For several viruses, including a ssDNA parvovirus, dsRNA has been detected by immunofluorescence analysis (Weber et al. 2006, https://doi.org/10.1128/JVI.80.10.5059-5064.2006; Son et al. 2015, https://doi.org/10.1128/JVI.01299-15). For ssRNA viruses, dsRNA in an intermediate generated in genome replication, while for DNA viruses, it is assumed that convergent transcription from bidirectional promoters results in the formation of overlapping RNAs (Bronkhorst et al. 2012, www.pnas.org/cgi/doi/10.1073/pnas.1207213109). Following the reviewer’s comment, the sentence at lines 811-813 in the manuscript was slightly modified and two references [74,75] added.
Reviewer 3: Is there a possibility that the possible ssDNA virus is a fragment that is inserted into the genome of the fungus?
Authors: We do not know. Further characterization of this and other selected viruses is in progress.
Reviewer 3: As this finding is somewhat surprising, it would be convenient to ensure that it is really this virus and not some artifact or contamination. Is it possible to sequence or detect the entire genome of the ssDNA virus? For example: Detect the size of the entire genome using a Southern blot experiment using the region determined by PCR as a probe.
This type of virus from the Parvoviridae family has a length between 4 and 6 kb. If the virus is inside M. fruticola, its genome must be full lenght.
Authors: Thank you for the valuable suggestion. To exclude assembly artifacts or contaminations, total DNAs from single isolates of the pool 4 of Serbian isolates were extracted and analysed by PCR and Sanger sequencing. A PCR primer pair designed on the two short contigs (300-400 nt) initially assembled and in accordance with the genomic structure of parvoviruses allowed us to reconstruct a 1,617-nt fragment of the parvovirus (Lines 709-712). PCR validation was extended by using additional specific primers designed on the reconstructed sequence. Both metagenomics and PCR analysis suggest the presence of the parvovirus at very low density in the fungal, hampering Southern blot experiments. The aim of this work was to explore the mycovirome of M. fructicola. Further research is in progress for a deeper characterization of the parvovirus and other selected viruses, including the obtainment of complete genomic sequences and investigations of possible role in fungal biology (Lines 1254-1255).
Reviewer 3: In this sentence:
Ln 719 “dsRNA analysis permits indeed detection of viruses with dsRNA as well as ssRNA or DNA genomes due to generation of abundant transient dsRNA intermediates during virus replication” REF [48].
I guess reference 48 is not related to this assertion. Please confirm that this sentence is related to this reference.
Authors: The suggestion has been accepted and ref [48] removed. The sentence was modified and two references [74,75] were added as above specified (Lines 811-813).
Reviewer 3: A figure to clarify the way in which dsRNA sequencing has been attempted and a diagram of some virus from the Parvoviridae family would be appreciated.
Authors: We sequenced dsRNA extracted and purified by CF11 cellulose chromatography from pools of fungal isolates, as described in materials and methods and schematized in subsections. A reference to the ICTV Report chapter describing Parvoviridae (Cotmore et al. 2019) has been added to the manuscript (Lines 714). The workflow was relatively simple and we believe useless to add further Figures to the manuscript.
Reviewer 3: Is there a phenotype associated with the isolates that present this ssDNA virus?
Authors: The isolate Mfr407 in which the ssDNA virus was detected did not show any visible alteration of colony morphology, but we cannot exclude other phenotypic effects due to viral infection. Further characterization of the parvovirus and other selected viruses is in progress.
Reviewer 3: Are there different phenotypes in all isolates used in this article?
Authors: As expected, there was a broad phenotypic variability among the M. fructicola isolates which in any cases would overlap the variation potentially caused by viruses. Further characterization of selected viruses including their potential effects on phenotype and virulence is in progress. This will be done through comparison between infected and “cured” virus-free isolates.
Reviewer 3: The possible origin of this possible virus in the fungus is not explained. Therefore, it might be interesting to discuss the possible origin of this novel virus in fungi
Authors: We do not have enough information to speculate about the possible origin of this virus in fungi. The host spectrum of members of the Parvoviridae family is very broad and includes humans, mammals, fishes, birds, tunicates, arthropods and flatworms. The characterization of the full-length genome of the virus, the study on its phylogenetic relationships with parvoviruses from different hosts, and on possible transmission modes is surly worthwhile of further research.
Reviewer 3: In other families, it would be good to include at least a diagram of the genome of the most representative viruses found.
Authors: Number and size of ORFs in the assembled genomes are reported in Table 4. We believe additional diagrams of the genome in the Figures would not be more informative since most are very simple.
Reviewer 3: About Binarnavirus: Recently a new article by Sato, Y. et al. was published. 'A new tetra-segmented splipalmivirus with divided RdRP domains from Cryphonectria naterciae, a fungus found on chestnut and cork oak trees in Europe'. Virus Res. 307, 198606 (2022).
They reported in C. naterciea a positive-sense (+), single-stranded (ss) RNA viruses with divided RNA-dependent RNA polymerase (RdRP) domains. These viruses are termed splipalmiviruses or polynarnaviruses and I think that will be a better name instead of binarnavirus because CnSpV1 has a tetra-segmented genome. A (+)ssRNA genome (RNA1 to RNA4). The RdRP domain is separately encoded by RNA1 (motifs F, A, and B) and RNA2 (motifs C and D). RNA3 encoded a hypothetical protein shows similarity to the conserved counterpart in some splipalmiviruses. And the RNA4 encoded protein does not show similarity to known proteins. Therefore, did you find RNA2, RNA3, and RNA4 in your dsRNA-seq?
Possibly it would be more correct to change Monilinia fructicola binarnavirus 1 name by another name such as splipalmiviruses or polynarnaviruses. Please check Sato, Y. et al. to confirm which narnavirus is Monilinia fructicola binarnavirus 1 according to its segments.
Authors: The suggestion was accepted and the name of Monilinia fructicola binarnavirus 1 was changed to Monilinia fructicola splipalmivirus 1 (MfrcSPV1). A partial RNA2 segment of 2,123 nt of the virus showing 47.4% aa sequence identity with the hypothetical protein on the RNA2 segment of Botrytis cinerea binarnavirus 2 was identified and included in the results and discussion sections (Lines 285-296 and 1018-1021, Table 4). However, we were not able to detect further putative RNA segments by homology search using conserved regions of the available sequences of Magnaporthe oryzae narnavirus 1, Aspergillus fumigatus narnavirus 2 and Cryphonectria naterciae splipalmivirus 1.
Minor critiques
Reviewer 3: I found an error in figure number 7. I guess panel G is for Barnaviridae and panel H is for Parvoviridae.
Authors: Thanks to the referee for the careful reading of the manuscript. We have corrected the error by modifying the figure legend (Line 801).
Reviewer: In all phylogenetic trees the Bootstrap % are not easy to read.
Authors: The suggestion has been accepted and in the new version of the paper, Figures 1 to 6 were modified to make bigger the font of Bootstrap values.
Reviewer 3: I miss all the accession numbers for new viruses and for viruses used in phylogenetic trees. It would be great to include a new table with all accession lists and all characteristics of the virus annotated (e.g., length, proteins, segment, etc).
Authors: Accession numbers of all viral sequences used for phylogenetic analyses are detailed in Supplementary Table S1. Furthermore, accession numbers of raw sequencing data and those of new viruses identified during this work are now provided in the ‘Data Availability Statement’.
Conclusions
Overall, I think the article is interesting in the field and can provide information to the scientific community working on this phytopathogenic fungus. Although the article does not present major findings, there is a comprehensive description of Monilinea fruticola mycovirome. In any case, I think the article could be improved. One of the aspects that should be improved is regarding the finding of the ssDNA virus. I guess it is necessary to confirm that it is really this complete virus inside the fungus and that it is not an artifact, or a sequence inserted in the fungal genome. For such issue, I recommend a major revision to confirm this ssDNA virus is replicating inside Monilinea fruticola cell. I also think it would be interesting to try to connect the presence of the virus with some aspect of the phenotype (e.g., virulence). And finally, it would be great to confirm the multisegmented genome of the narnavirus Monilinia fructicola binarnavirus 1.
Round 2
Reviewer 1 Report
In my opion, it will be accepted in present form.
Reviewer 3 Report
I would like to thank the authors for the effort to respond to most of my improvement tips. In my opinion, the authors have improved the article and, therefore, I recommend its publication.
Thanks